fluid mechanics/biomedical engineering/nanotechnology

Janus microdimer, magnetic actuation, numerical simulation, Stokes flow, swimming

**Author for correspondence:**
Jinyou Yang
e-mail: jyyang@cmu.edu.cn

# Janus microdimer swimming in an oscillating magnetic field

## Jinyou Yang

School of Fundamental Sciences, China Medical University, Shenyang 110122, People's Republic of China

 JY, 0000-0001-6256-1899

Artificial microswimmers powered by magnetic fields have numerous applications, such as drug delivery, biosensing for minimally invasive medicine and environmental remediation. Recently, a Janus microdimer surface walker that can be propelled by an oscillating magnetic field near a surface was reported by Li *et al.* (*Adv. Funct. Mater.* **28**, 1706066. (doi:10.1002/adfm.201706066)). To clarify the mechanism for the surface walker, we numerically studied in detail a Janus microdimer swimming near a wall actuated by an oscillating magnetic field. The results showed that a Janus microdimer in an oscillating magnetic field can produce magnetic torque in the *y*-direction, which eventually propels the Janus microdimer along the *x*-direction near a wall. Furthermore, we found that the Janus microdimer can also move along a special direction in an oscillating magnetic field with two orientations without a wall. The knowledge obtained in this study is fundamental for understanding the interactions between a Janus microdimer and surfaces in an oscillating magnetic field and is useful for controlling Janus microdimer motion with or without a wall.

## 1. Introduction

In 1977, Purcell [1] proposed the famous theorem that the locomotion of microorganisms has to perform a non-reciprocal periodic motion to enable propulsion at low Reynolds numbers, where drag dominates over inertia. Non-reciprocal means that the time-reversed motion is not the same as the original motion [2]. Natural microorganisms exploit anisotropic drag to break time reversibility. Understanding propulsion strategies at low Reynolds numbers is useful to establish artificial microswimmers by using field activation [3,4].

The artificial microswimmer is a cutting-edge technology due to its high potential for applications in drug delivery, biosensing to minimally invasive medicine and environmental remediation [3–6]. For example, Wu *et al.* [7,8] reported a detailed investigation of the migration of micromotors towards targeted regions. Within

**Figure 1.** Schematic diagram of the Janus microdimer model in the magnetic field and the problem settings. $a_s$ is the radius of a Janus microsphere coated by a permanent magnetic material (blue part) and $a$ is the radius of the other part of the Janus particle. A flat wall exists at $z = 0$; **m** is the magnetic moment of the Janus particle, $\mathbf{x}_c$ is the geometric centre and $\mathbf{x}_b$ is the magnetic centre of the Janus particle. The angle $\alpha$ lies between the major axis of the dimer and the magnetic field direction. This angle is approximately 46°, similar to the experimental result (Li *et al.* [10]).

the last decade, various methods have been proposed for powering these microswimmers [3,4,9]. Among these powering methods, magnetic fields have been used to 'fuel' such devices, which is particularly promising for remotely and non-invasively actuating of microswimmers in a liquid environment [10–12].

Several mechanisms for magnetic actuation have been proposed based on the magnetic field application mode. First, magnetic particles can be actuated by coupling hydrodynamic torque and magnetic torque by using a static uniform magnetic field to pin the orientation of particles [13,14]. Second, magnetic microswimmers can be actuated by rotating or bending their body in a rotating magnetic field, or actuated by a precessing magnetic field to move along a surface [5,15–17]. Third, microswimmers behave as objects that can change shape during one period of the oscillating magnetic field to achieve migration in Stokes flow [2,6,10,12].

Recently, a microswimmer called a 'surface walker' was actuated by a magnetic field to move near a surface [10,17]. For example, Li *et al.* [10] conducted a Janus microdimer surface walker investigation in an oscillating magnetic field, which requires little sophisticated Helmholtz set-ups. These studies are important in many proposed applications such as blood vessels and microfluid chips. Therefore, we present a thorough investigation of a Janus microdimer that was constructed by Li *et al.* [10] in an oscillating magnetic field to clarify the mechanism of a Janus microdimer surface walk on a wall. We study the influence of the magnetic field gradient on the swimming speed and motion direction and demonstrate that the Janus microdimer can take on different directions via a change in the density of particles. We found the Janus microdimer can also move along a special direction in two orientated oscillating magnetic fields without a wall.

The paper is organized as follows. In §2, we explain the problem settings, basic equations and numerical methods. The effect of the wall, magnetic field gradient and particle density on the Janus microdimer surface walker is investigated in an oscillating magnetic field with one orientation in §3. In §4, we impose an oscillating magnetic field with two orientations without a wall and discuss its effect. We present our conclusions in §5.

## 2. Material and methods

Hereafter, all quantities are non-dimensionalized using characteristic length $a$, Boltzmann constant $k_b$, viscosity $\eta$ and thermodynamic temperature $T$, where $a$ is the radius of a Janus particle. The symbol * represents a dimensionless quantity; physical quantities without * represent a dimensional quantity.

### 2.1. Problem settings

The behaviour of a Janus microdimer in an oscillating magnetic field near a wall was investigated in this study. A flat wall is located at $z = 0$ in a rectangular coordinate system. Figure 1 shows a schematic

diagram of the problem settings. We consider a Janus microsphere coated by a permanent magnetic material in a Newtonian fluid of viscosity $\eta$ and density $\rho_{liquid}$. Therefore, the microdimer interacts with the surrounding fluid via hydrodynamic and magnetic interaction. The Janus particle has a magnetic dipole moment [2,18,19]

$$\mathbf{m} = \frac{2\pi(a_s^3 - a^3)}{3\mu_0}\chi\mathbf{B}_{initial},\tag{2.1}$$

where $a_s$ is the radius of the Janus microsphere coated by a permanent magnetic material and $a$ is the radius of the other part of the Janus particle, $\mu_0 = 4\pi \times 10^{-7}\,\text{N/A}^2$ is the permeability of free space, $\chi$ is the magnetic susceptibility of the permanent magnetic material, $\mathbf{B}_{initial}$ is the external magnetic induction in the initial condition. The external oscillating magnetic field was set to be the same as that in the study of Li et al. [10]

$$\mathbf{H} = \mathbf{B}_{initial} * \exp(-k_{exb} \cdot k_0 \cdot x) * \sin(f \cdot t)/\mu_0,\tag{2.2}$$

where $k_{exb} \cdot k_0$ shows the magnetic field as a function of the separation distance, $k_0$ is a parameter from the study of Li et al. [10], $f$ is the frequency of the external oscillating magnetic field, while $k_{exb}$ was varied in this study.

The magnetic force and torque for the Janus particle was calculated from the dipole–dipole interaction energy and the energy due to the interaction between the Janus particle and the applied magnetic field.

The dipole–dipole interaction energy of particle $i$ provoked by particle $j$ is calculated by [2,18–21]

$$U_{i-j}^m = \frac{\mu_0}{4\pi}\left[\frac{\mathbf{m}_i \cdot \mathbf{m}_j}{r_{ij}^3} - \frac{(\mathbf{m}_i \cdot \mathbf{r}_{ij})(\mathbf{m}_j \cdot \mathbf{r}_{ij})}{r_{ij}^5}\right].\tag{2.3}$$

The energy due to interaction between the Janus particle and the applied magnetic field is calculated by

$$U_i^H = \mathbf{m}_i \cdot \mathbf{H},\tag{2.4}$$

where $\mathbf{r}_{ij} = \mathbf{x}_{bi} - \mathbf{x}_{bj}$, $\mathbf{x}_{bi}$ is the magnetic moment centre of the $i$th particle.

A non-dimensional form for the magnetic force and torque can be derived directly from the energy by

$$\mathbf{F}_{i-j}^{m*} = -\frac{a}{k_b T}\nabla_{r_{ij}}U_{i-j}^m = R_m\frac{1}{(r_{ij})^4}[(\mathbf{n}_i \cdot \mathbf{n}_j)\hat{\mathbf{r}}_{ij} - 5(\mathbf{n}_i \cdot \hat{\mathbf{r}}_{ij})(\mathbf{n}_j \cdot \hat{\mathbf{r}}_{ij})\hat{\mathbf{r}}_{ij} + \{(\mathbf{n}_j \cdot \hat{\mathbf{r}}_{ij})\mathbf{n}_i + (\mathbf{n}_i \cdot \hat{\mathbf{r}}_{ij})\mathbf{n}_j\}],\tag{2.5}$$

$$\mathbf{F}_i^{H*} = \frac{a}{k_b T}\mu_0\,(\mathbf{m}_i \cdot \nabla)\mathbf{H},\tag{2.6}$$

$$\mathbf{T}_{i-j}^{m*} = -\frac{1}{k_b T}\,\mathbf{m}_i \times \nabla_{\mathbf{m}_i}U_{i-j}^m = R_m\frac{1}{3(r_{ij})^3}[(\mathbf{n}_i \times \mathbf{n}_j) - 3(\mathbf{n}_i \times \hat{\mathbf{r}}_{ij})(\mathbf{n}_j \cdot \hat{\mathbf{r}}_{ij})]\tag{2.7}$$

and

$$\mathbf{T}_i^{H*} = \frac{\mu_0}{k_b T}\mathbf{m}_i \times \mathbf{H},\tag{2.8}$$

where $\mathbf{n}_i$ is the unit vector of the magnetic moment of the $i$th Janus particle, $R_m = 3\mu_0 m_i m_j/4\pi k_b T a^3$.

The coating on a Janus particle can make a magnetic dipole moment shift from the geometric centre of the particle [22]. This dipole offset leads to a torque on particle $i$ due to the magnetic dipole–dipole interaction force,

$$\mathbf{T}_i^{\mathbf{F}_{i-j}^m *} = \frac{(\mathbf{x}_{bi} - \mathbf{x}_{ci})}{a} \times \mathbf{F}_{i-j}^{m*}.\tag{2.9}$$

In this study, a Derjaguin–Landau–Verwey–Overbeek (DLVO)-type short-range repulsive force was calculated for the repulsive nature of particle–particle and particle–wall interactions, which is given by [23]

$$\mathbf{F}_{i-j}^{rep*} = \alpha_1 \frac{\exp(-\alpha_2 r_{c-ij})}{1 - \exp(-\alpha_2 r_{c-ij})}\hat{\mathbf{r}}_{c-ij}\tag{2.10}$$

and

$$\mathbf{F}_{i-wall}^{rep*} = \alpha_3 \frac{\exp(-\alpha_4\varepsilon_{min})}{1 - \exp(-\alpha_4\varepsilon_{min})}\mathbf{d},\tag{2.11}$$

where $\alpha_1$ and $\alpha_3$ are coefficients that control the magnitude of the force, $\alpha_2$ and $\alpha_4$ are coefficients that control the decay length, $r_{c-ij}$ is the minimum distance of the Janus particle surface, $\varepsilon_{min}$ is the minimum distance

between the Janus particle surface and the wall and $d$ is the unit vector connecting the minimum separation point from the wall.

## 2.2. Basic equations

Due to the small size of the Janus particle, we neglect inertial effects in the flow field and assume Stokes flow. A boundary element method was used as in our former study [24]. In the Stokes flow regime, the velocity around the Janus microdimer can be determined by the following boundary integral equation [25]:

$$u_i^*(\mathbf{x}) - u_i^{\infty*}(\mathbf{x}) = -\frac{1}{8\pi}\int_{\text{particle}} G_{ij}^w(\mathbf{x} - \mathbf{y})t_j(\mathbf{y})\,\mathrm{d}A_c, \tag{2.12}$$

where $\mathbf{u}^*(\mathbf{x})$ is the velocity at position $\mathbf{x}$, $\mathbf{u}^{\infty*}(\mathbf{x})$ is the background velocity, $A_c$ is the surface of the Janus microdimer and $\mathbf{t}$ is the traction force. $\mathbf{G}^w$ is Green's function for the half space bounded by a no-slip wall, which is given by [26]

$$G_{ij}^w(\mathbf{x} - \mathbf{y}) = \left(\frac{\delta_{ij}}{r} + \frac{r_i r_j}{r^3}\right) - \left(\frac{\delta_{ij}}{R} + \frac{R_i R_j}{R^3}\right) + 2h(\delta_{j\alpha}\delta_{\alpha k} - \delta_{j3}\delta_{3k})\frac{\partial}{\partial R_k}\left\{\frac{hR_i}{R^3} - \left(\frac{\delta_{i3}}{R} + \frac{R_i R_3}{R^3}\right)\right\}, \tag{2.13}$$

where $\mathbf{y} = (y_1, y_2, h)$, $r = [(x_1 - y_1)^2 + (x_2 - y_2)^2 + (x_3 - h)^2]^{1/2}$, $R = [(x_1 - y_1)^2 + (x_2 - y_2)^2 + (x_3 + h)^2]^{1/2}$ and $\alpha = 1, 2$.

In §4, we investigate the Janus microdimer motion propelled by two orientations of an oscillating magnetic field on the $x$–$z$ plane and $y$–$z$ plane without wall. Therefore, Green's function is given by $G_{ij}(\mathbf{x} - \mathbf{y}) = (\delta_{ij}/r + r_i r_j/r^3)$.

Similar to the approach of Li et al. [10], the Janus microdimers were silica microspheres with a diameter of 3 µm half-coated with a 15 nm thick layer of nickel. The composition of gravity and buoyant force are given by

$$\mathbf{F}^{G*} = \frac{g}{k_b T/a}\left[\frac{4\pi a^3}{3}(\rho_{\text{silica}} - \rho_{\text{liquid}}) + \frac{2\pi(a_s^3 - a^3)}{3}(\rho_{\text{nickel}} - \rho_{\text{liquid}})\right], \tag{2.14}$$

where $\rho_{\text{silica}}$ is the density of the silica microsphere, $\rho_{\text{nickel}}$ is the density of the nickel layer, $g$ is the acceleration due to gravity. Therefore, the centre of buoyancy of the Janus particle ($\mathbf{x}_{Gi}$) is offset from its geometric centre. A torque $\mathbf{T}_i^G = (\mathbf{x}_{Gi} - \mathbf{x}_{ci}) \times \mathbf{F}^G$ on the particle was considered.

As the system is force- and torque-free, the force and torque equilibrium equations are given by

$$\left.\begin{aligned}
\mathbf{F}_i^* &= \int_{\text{particle}} \mathbf{t}(\mathbf{x})\,\mathrm{d}A_c + \sum_{j\neq i}^n [\mathbf{F}_{i-j}^{m*} + \mathbf{F}_{i-j}^{\text{rep}*}] + \mathbf{F}_i^{H*} + \mathbf{F}_{i-\text{wall}}^{\text{rep}*} + \mathbf{F}_i^{G*} = 0 \\
\text{and} \quad \mathbf{T}_i^* &= \int_{\text{particle}} (\mathbf{x} - \mathbf{x}_{ci}) \times \mathbf{t}(\mathbf{x})\,\mathrm{d}A_c + \sum_{j\neq i}^n [\mathbf{T}_{i-j}^{m*} + \mathbf{T}_i^{\mathbf{F}_{i-j}^{m*}}] + \mathbf{T}_i^{H*} + \mathbf{T}_i^{G*} = 0,
\end{aligned}\right\} \tag{2.15}$$

where the integral is over the whole Janus particle surface.

## 2.3. Numerical methods

The boundary element method was employed to solve for the Janus microdimer motion. In total, 320 elements were generated on the Janus particle.

The surface integral in the basic equations was performed on a triangular element using 28-point Gaussian polynomials, and the singularity in the integration was solved analytically [27]. Time marching was performed using the fourth-order Adams–Bashforth method.

The parameters used in the present study are listed in table 1. The parameters used in this study are similar to the study of Li et al. [10]. As illustrated in figure 1, the magnetic moment centre ($\mathbf{x}_{bi}$) was checked for different conditions; when $\mathbf{x}_{bi} = \mathbf{x}_{ci} - (0,0,0.5a)$, the angle $\alpha$ between the major axis of the dimer and the magnetic field direction is similar to the experimental result of Li et al. [10]. In this study, we found that the coefficients that control the magnitude of the force ($\alpha_1$, $\alpha_3$) and the coefficients that control the decay length ($\alpha_2$, $\alpha_4$) affect the motion speed of the Janus microdimer. These parameters were selected in table 1 to ensure the result of case ($B_{\text{initial}} = 2.7$ mT, $f = 20$ Hz) was similar to the experimental result of Li et al. [10]. The particle density ($\rho$) was calculated from the material composition of the particle.

**Table 1.** Parameters used in this study.

| | | |
|---|---|---|
| radius of Janus particle $a$ | 3 µm | Li *et al.* [10] |
| radius of Janus particle with nickels layer $a_s$ | 3 + 15 µm | Li *et al.* [10] |
| Boltzmann constant $k_b$ | $1.380649 \times 10^{-23}$ J K$^{-1}$ | |
| density of silica $\rho_{silica}$ | $2 \times 10^3$ kg m$^{-3}$ | |
| density of nickel layer $\rho_{nickel}$ | $8.9 \times 10^3$ kg m$^{-3}$ | |
| density of liquid $\rho_{liquid}$ | $10^3$ kg m$^{-3}$ | |
| magnetic susceptibility of nickel $\chi$ | 600 | |
| thermodynamic temperature $T$ | 300 K | |
| magnetic field as a function of the separation distance $k_{exb} \cdot k_0$ | $k_{exb} = -2, -1, 0, 1, 2$ | Li *et al.* [10] |
| | $k_0 = 1/1.145$ cm$^{-1}$ | |
| initial external magnetic induction $B_{initial}$ | 2.7, 5.6 (mT) | Li *et al.* [10] |
| coefficient to control the magnitude of the force $\alpha_1$ and $\alpha_3$ | $\alpha_1 = 1 \times 10^4$ | |
| | $\alpha_3 = 2 \times 10^4$ | |
| coefficient to control the decay length $\alpha_2$ and $\alpha_4$ | $\alpha_2 = 2.2$ | |
| | $\alpha_4 = 20$ | |

# 3. Study of a Janus microdimer surface walker in an oscillating magnetic field

A Janus microdimer surface walker in an oscillating magnetic field with one orientation with a wall was investigated.

## 3.1. Wall-induced Janus microdimer surface walk

When the Janus microdimer reached a steady state in a uniform magnetic field ($B_{initial}$) due to the magnetic dipole moment shift from the geometric centre of the particle, and the particles line as indicated in figure 1. We note that the offset of the magnetic dipole moment, as well as the repulsive force between particles, affects not only the angle between the major axis of the dimer and the magnetic field direction but also whether the Janus microdimer can walk on the surface.

The different phases of the Janus microdimer in one period of motion are shown in figure 2. After directional changes in the external magnetic field, the Janus microdimer rolled to move a displacement in the direction of the external magnetic field, as shown in figure 2*a*. During the first one-fourth period of the sinusoidal cycle, one of the two particles in the dimer rolled clockwise a little upward (*z*-direction) and then forward, while on the other hand, the other particle rolled along the same direction but slid backward, as shown in figure 2*b*. This backward sliding particle was propelled by the resistance due to the wall. In the next half of the cycle, the two particles switched roles, and the trailing particle rolled forward to become the leading particle, which was then repeated in subsequent cycles, propelling the Janus microdimer forward as a surface walker.

The magnetic torque on the Janus microdimer was investigated. Figure 3*a* shows the magnetic torque on a Janus microdimer in the *x*-direction; 'instantaneous' denotes the magnetic torque time changes during one period; 'time-average' is the averaged magnetic torque over time, for which torque increases negatively in the *x*-direction, and then decreases to zero at the end of period. Figure 3*b* shows the magnetic torque on the Janus microdimer in the *y*-direction. The Janus microdimer rotates clockwise around the *y*-axis due to the magnetic torque in the *y*-direction; it is acted upon by a net force in the *x*-direction, induced by the presence of the wall [28]. The microdimer is then pushed along the *x*-direction on the surface. Figure 3*c* shows the magnetic torque on the Janus microdimer in the *z*-direction. The time-averaged *z*-direction magnetic torque is approximately zero during one period.

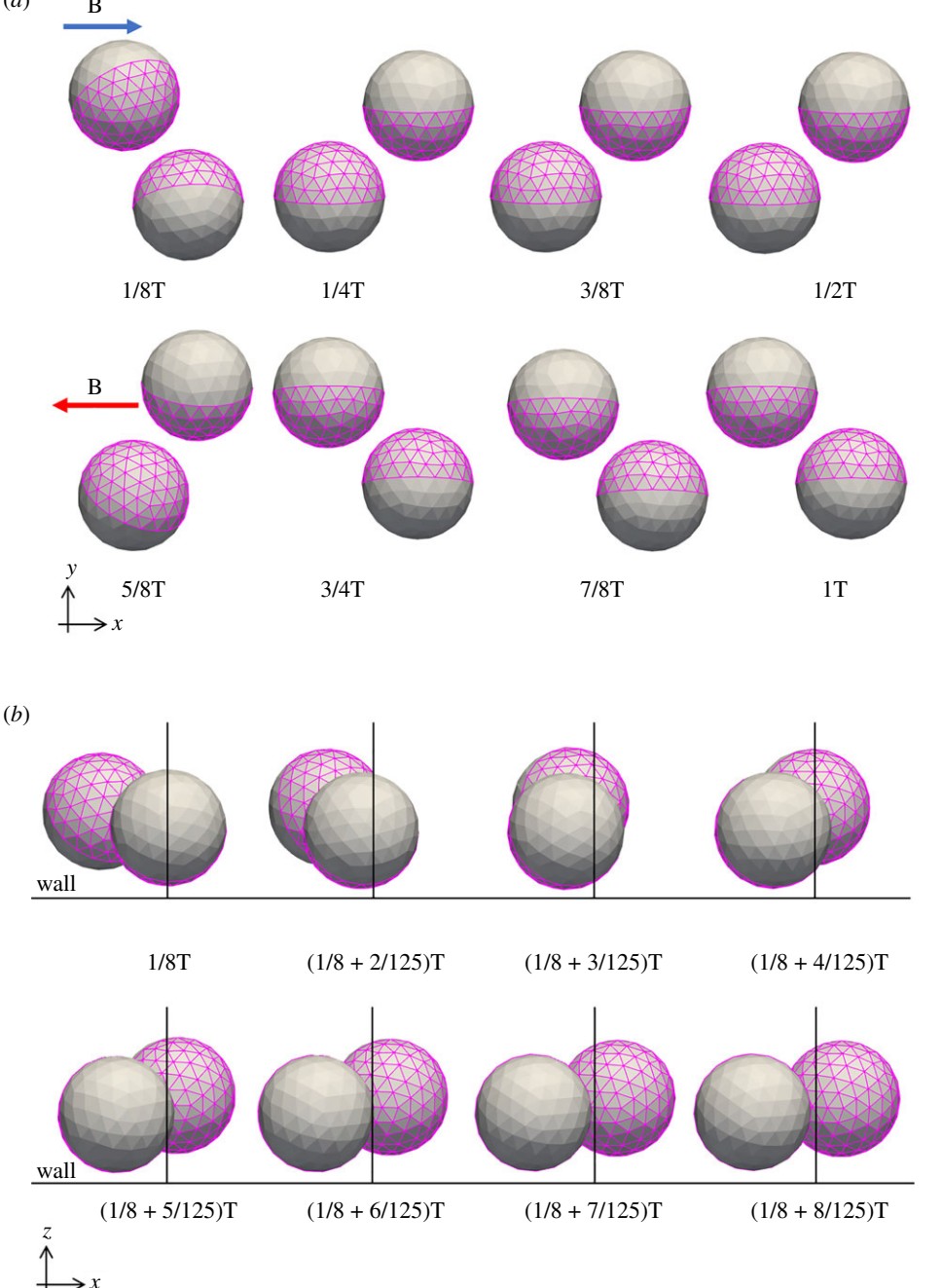

**Figure 2.** The different phases of a Janus microdimer in one period of motion under the condition of an external magnetic field at 20 Hz, 2.7 mT. (*a*) Snapshot in one period on the *xy* plane. (*b*) Snapshot from (1/8) period to (1/8 + 8/125) period on the *xz* plane.

These results illustrate that a Janus microdimer near a wall in an oscillating magnetic field can produce a magnetic torque in the *y*-direction, which eventually propels the Janus microdimer along the *x*-direction near the wall.

## 3.2. Effect of magnetic field gradient ($k_{exb}$) on Janus microdimer motion

The effect of the gradient of an external magnetic field on the Janus microdimer motion on the wall was investigated. Similar to the experimental parameter of Li *et al*. [10], $k_{exb} \cdot k_0$ in equation (2.2) was used to express the magnetic field as a function of the separation distance. When $k_{exb} = 1$, the parameter of the magnetic field gradient is the same as the experimental parameter of Li *et al*. [10]. Figure 4

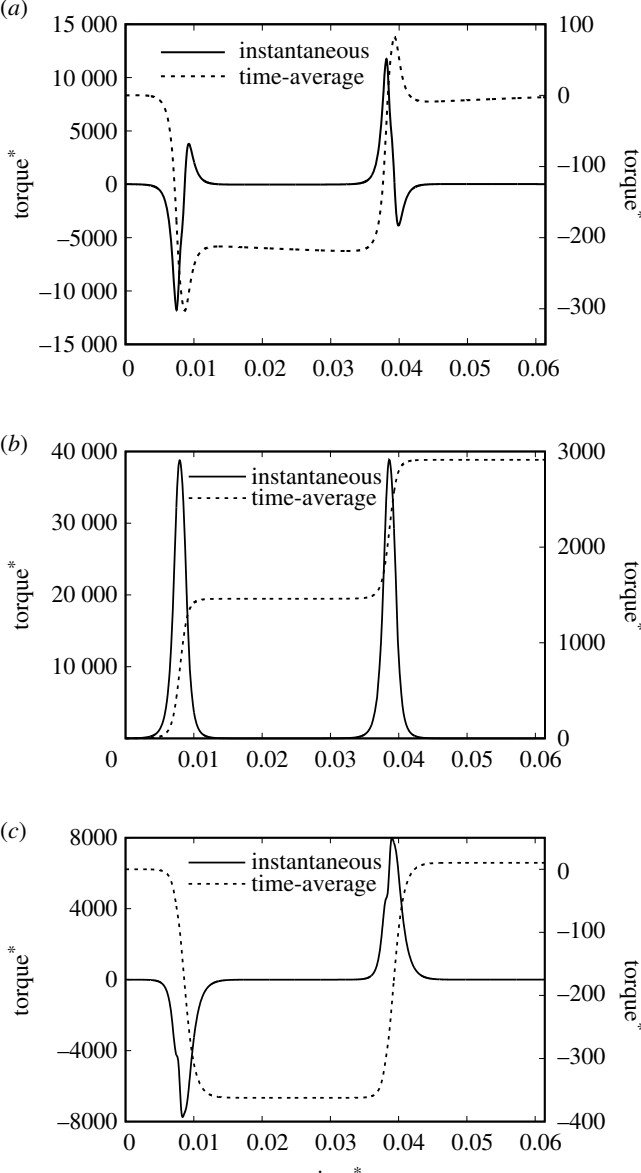

**Figure 3.** The magnetic torque on the Janus microdimer during one period under the condition of an external magnetic field at 20 Hz, 2.7 mT. 'instantaneous' is the time change of the torque, 'time-average' is the time averaged torque. (*a*) Torque in the *x*-direction. (*b*) Torque in the *y*-direction. (*c*) Torque in the *z*-direction.

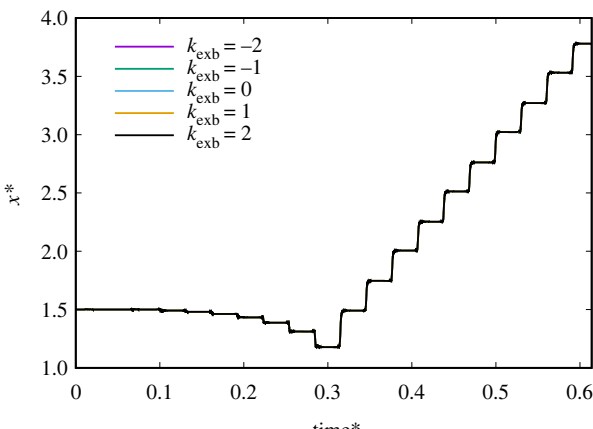

**Figure 4.** Time change for Janus microdimer motion in the *x*-direction on different magnetic field gradients ($k_{exb}$) during 10 periods under the condition of an external magnetic field at 20 Hz, 2.7 mT.

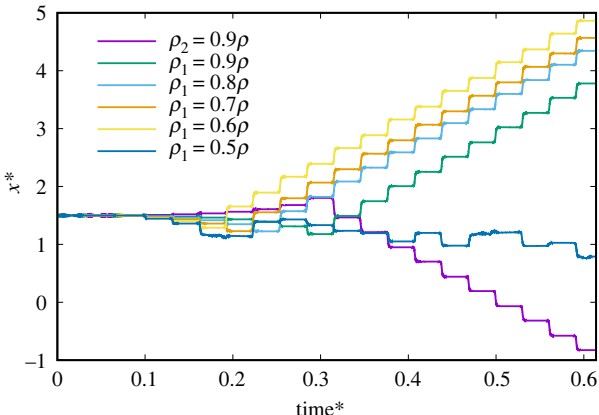

**Figure 5.** Time change of Janus microdimer motion in $x$-direction for different particle densities (unchanged magnetic dipole moment) during 10 periods under the condition of an external magnetic field at 20 Hz, 2.7 mT. $\rho_1 = 0.9\rho$ means the density of the first particle $\rho_1$ is $0.9\rho$ and the density of the second particle is $\rho$, where $\rho$ is the basic particle density that is the same as the experimental parameter (Li *et al*. [10]).

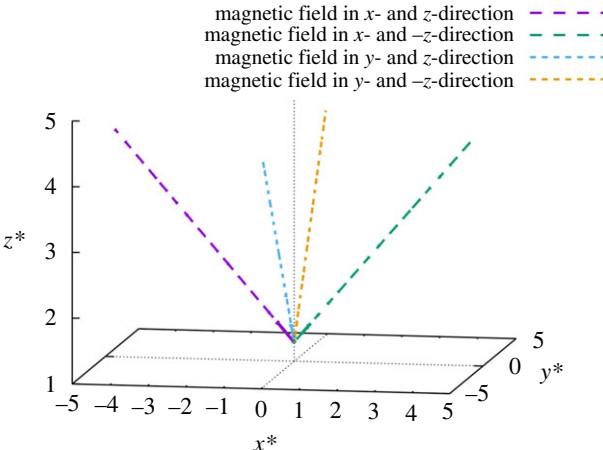

**Figure 6.** Trajectories for Janus microdimer motion in different two orientations for an oscillating magnetic field under the condition of $f = 20$ Hz and $k_{exb} = 1$ during 10 periods. The initial external magnetic induction ($\boldsymbol{B}_{initial}$) in the $x$- and $z$-directions is 5.6 and 8.4 mT, respectively. The initial magnetic field orientation in the positive $x$- and $z$-directions produced Janus microdimer motion along the negative $x$-direction and positive $z$-direction on the $x$–$z$ plane. The initial magnetic field orientation in the positive $x$- and negative $z$-directions produced Janus microdimer motion along the positive $x$- and $z$-directions on the $x$–$z$ plane. The initial magnetic field orientation in the positive $y$- and $z$-directions produced Janus microdimer motion along the negative $y$-direction and positive $z$-direction on the $y$–$z$ plane. The initial magnetic field orientation in the positive $y$- and negative $z$-direction produced Janus microdimer motion along the positive $y$-direction and positive $z$-direction on the $y$–$z$ plane.

shows the effect of magnetic field gradient on the motion speed and direction of the Janus microdimer under the condition of an external magnetic field at 20 Hz, 2.7 mT. These results indicate that there is no effect on the motion speed and direction of the Janus microdimer due to the magnetic field gradient.

## 3.3. Effect of particle density on Janus microdimer motion

Figure 5 shows the effect of particle density $\rho_i$ on the time change of the position in the $x$-direction under the conditions of an external magnetic field at 20 Hz, 2.7 mT. We checked different densities (0.5–0.9$\rho$) for the first particle with constant density ($\rho_2 = \rho$) for the second particle, for which $\rho$ is the basic particle density that is the same as the experimental parameter of Li *et al*. [10]. These results illustrate that the particle density plays an important role in the motion direction of a Janus microdimer. In addition, when the density difference is not greater than half, the speed of the Janus microdimer motion is not affected; otherwise, the Janus microdimer does not move on the wall.

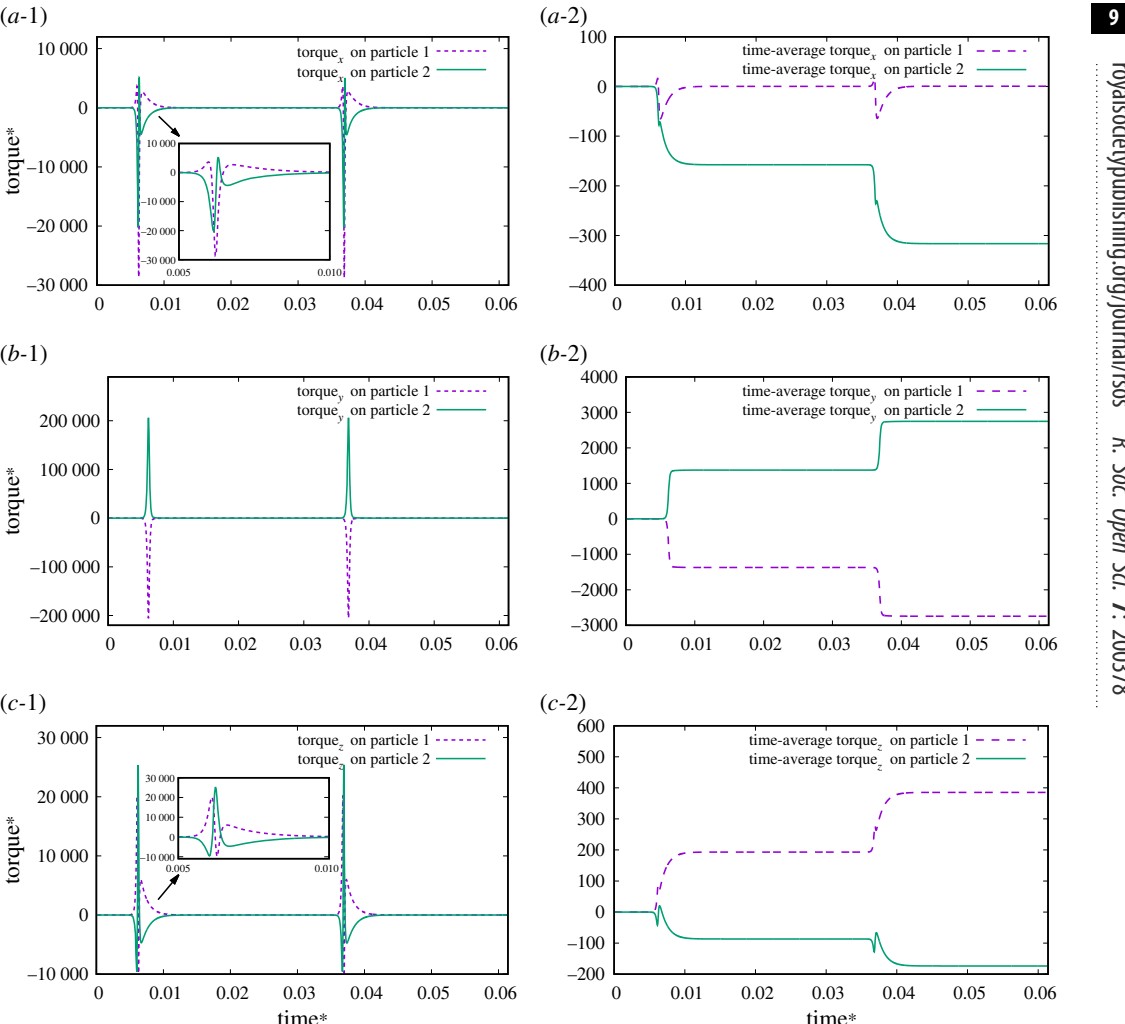

**Figure 7.** The magnetic torque on a Janus microdimer without a wall during one period under the condition of an external magnetic field at 20 Hz, 5.6 mT in the $x$-direction and 8.4 mT in the $z$-direction. ($a$-1) Torque in the $x$-direction on the first and second particles. ($a$-2) Time-averaged torque in the $x$-direction on the first and second particles. ($b$-1) Torque in the $y$-direction on the first and second particles. ($a$-2) Time-averaged torque in the $y$-direction on the first and second particles. ($c$-1) Torque in the $z$-direction on the first and second particles. ($c$-2) Time-averaged torque in the $z$-direction on the first and second particles.

# 4. Janus microdimer motion propelled by an oscillating magnetic field with two orientations without a wall

We investigated Janus microdimer motion propelled by two orientations of an oscillating magnetic field on the $x$–$z$ plane and $y$–$z$ plane without a wall. The Janus microdimer trajectory results are shown in figure 6 with an external magnetic field condition of $f = 20$ Hz and $k_{exb} = 1$. The Janus microdimer moves along the negative $x$-direction and positive $z$-direction on the $x$–$z$ plane with the initial magnetic field orientation in the positive $x$- and $z$-directions without a wall. The initial external magnetic induction strength ($B_{initial}$) is different in the $x$-direction (5.6 mT) and $z$-direction (8.4 mT). The initial magnetic field orientation in the positive $x$- and negative $z$-direction was then changed, which produced Janus microdimer motion along the positive $x$- and $z$-directions on the $x$–$z$ plane. These results indicate that a Janus microdimer can be controlled to a target position on the $x$–$z$ or $y$–$z$ planes without a wall by using an oscillating magnetic field with two orientations.

Figure 7 shows the time changes and time-averaged magnetic torque in different directions for the particles under the condition of an external magnetic field of 20 Hz, 5.6 mT in the $x$-direction and 8.4 mT in the $z$-direction without a wall. An asymmetric torque in the $x$-direction and $z$-direction on

the first particle and second particle are shown in figure 7*a*-1 and *c*-1, and different torque intensities in the *x*- and *z*-directions on the first particle and second particle are shown in figure 7*a*-2 and *c*-2. Figure 7*b* shows a symmetric torque, with nearly no difference observed for the intensity of the torque in the *y*-direction. These results indicate that an oscillating magnetic field with two orientations exerts a torque along two coordinates to break time reversibility and produce a non-reciprocal motion pattern.

# 5. Conclusion

To clarify the detailed Janus microdimer surface walker studied by Li *et al.* [10] in experiments in an oscillating magnetic field, we numerically investigated the motion of a Janus microdimer near a flat wall. Our results illustrate that a Janus microdimer near a wall in an oscillating magnetic field can produce torque in the positive *y*-direction and can then be propelled along the *x*-direction.

We also applied an oscillating magnetic field with two orientations on the *x*–*z* and *y*–*z* planes without a wall to actuate a Janus microdimer along a special direction. The knowledge obtained in this study forms a fundamental basis for understanding the interactions between a Janus microdimer and surfaces in an oscillating magnetic field and is useful for controlling Janus microdimer motion with or without a wall.

Data accessibility. Data available from the Dryad Digital Repository: http://dx.doi.org/10.5061/dryad.p8cz8w9m7 [29].
Authors' contributions. J.Y. completed all the work for this paper.
Competing interests. I declare I have no competing interests.
Funding. I received no funding for this study.
Acknowledgements. Thanks Prof. Ishikawa Takuji for the suggestion of the numerical method and model.

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
