## [Reviewer comments · Royal Society Open Science]

Review History

RSOS-200378.R0 (Original submission)

Review form: Reviewer 1

Is the manuscript scientifically sound in its present form?

Yes

Are the interpretations and conclusions justified by the results?

Yes

Is the language acceptable?

Yes

Do you have any ethical concerns with this paper?

No

Have you any concerns about statistical analyses in this paper?

No

Recommendation?

Accept with minor revision (please list in comments)

Comments to the Author(s)

1. have you verified the numerical model? Is it enough for numerical simulation with 320 elements on the janus particle?
2. Have you verified experimentally the method of microrobots propelled by two orientation oscillating magnetic field without wall?
3. Please cite and discuss the latest work of microrobots in introduction, such as Science Robotics, 2019, 4 (32), eaax0613; Science advances, 2018, 4 (11), eaat4388; Nanomaterials 2019, 9 (12), 1672; Advanced Functional Materials, 2018, 28 (25), 1706066; Nano letters, 2017, 17 (8), 5092-5098; ACS nano, 2017, 11 (9), 9268-9275.
4. The English should be polished.

Review form: Reviewer 2**Is the manuscript scientifically sound in its present form?**

Yes

Are the interpretations and conclusions justified by the results?

Yes

Is the language acceptable?

Yes

Do you have any ethical concerns with this paper?

No

Have you any concerns about statistical analyses in this paper?

No

Recommendation?

Accept with minor revision (please list in comments)

Comments to the Author(s)

This is a fundamental research contribution that analyzes theoretically the motion of a Janus dimer system under oscillating magnetic fields, and the interactions between the dimer and surfaces. The paper is nicely explained. However, it is not clear to this author if the surfaces between the microdimers play also a role in this type of locomotion. Can the authors comment on this?

Decision letter (RSOS-200378.R0)

Dear Dr Yang,

On behalf of the Editors, we are pleased to inform you that your Manuscript RSOS-200378 "Janus microdimer swimming in oscillating magnetic field" has been accepted for publication in Royal Society Open Science subject to minor revision in accordance with the referees' reports. Please find the referees' comments along with any feedback from the Editors below my signature.

We invite you to respond to the comments and revise your manuscript. Below the referees' and Editors' comments (where applicable) we provide additional requirements. Final acceptance of

your manuscript is dependent on these requirements being met. We provide guidance below to help you prepare your revision.

Please submit your revised manuscript and required files (see below) no later than 7 days from today's (ie 10-Sep-2020) date. Note: the ScholarOne system will 'lock' if submission of the revision is attempted 7 or more days after the deadline. If you do not think you will be able to meet this deadline please contact the editorial office immediately.

on behalf of Professor Miles Padgett (Associate Editor) and Pietro Cicuta (Subject Editor)
openscience@royalsociety.org

Editorial Comments to Author:

As you have been requested to edit the written English, you must provide proof that you have done so: acceptable proof includes a certificate of language-editing from a language editing service or a signed letter from a native speaker of English. If you do not provide this proof, your manuscript may be returned to you.

For information about language editing services endorsed by the Royal Society, please follow the link below:

<https://royalsociety.org/journals/authors/language-polishing/>

Reviewer comments to Author:

Reviewer: 1
Comments to the Author(s)

1. have you verified the numerical model? Is it enough for numerical simulation with 320 elements on the janus particle?
2. Have you verified experimentally the method of microrobots propelled by two orientation oscillating magnetic field without wall?
3. Please cite and discuss the latest work of microrobots in introduction, such as Science Robotics, 2019, 4 (32), eaax0613; Science advances, 2018, 4 (11), eaat4388; Nanomaterials 2019, 9 (12), 1672; Advanced Functional Materials, 2018, 28 (25), 1706066; Nano letters, 2017, 17 (8), 5092-5098; ACS nano, 2017, 11 (9), 9268-9275.

4. The English should be polished.

Reviewer: 2

Comments to the Author(s)

This is a fundamental research contribution that analyzes theoretically the motion of a Janus dimer system under oscillating magnetic fields, and the interactions between the dimer and surfaces. The paper is nicely explained. However, it is not clear to this author if the surfaces between the microdimers play also a role in this type of locomotion. Can the authors comment on this?

===PREPARING YOUR MANUSCRIPT===

===PREPARING YOUR REVISION IN SCHOLARONE===

Please ensure that you include a summary of your paper at Step 2 'Type, Title, & Abstract'. This should be no more than 100 words to explain to a non-scientific audience the key findings of your

research. This will be included in a weekly highlights email circulated by the Royal Society press office to national UK, international, and scientific news outlets to promote your work.

-- Ensure that your data access statement meets the requirements at <https://royalsociety.org/journals/authors/author-guidelines/#data>. You should ensure that you cite the dataset in your reference list. If you have deposited data etc in the Dryad repository, please only include the 'For publication' link at this stage. You should remove the 'For review' link.

Author's Response to Decision Letter for (RSOS-200378.R0)

See Appendix A.

RSOS-200378.R1 (Revision)

Review form: Reviewer 1

Is the manuscript scientifically sound in its present form?

Yes

Are the interpretations and conclusions justified by the results?

Yes

Is the language acceptable?

Yes

Do you have any ethical concerns with this paper?

No

Have you any concerns about statistical analyses in this paper?

No

Recommendation?

Accept as is

Comments to the Author(s)

This Paper can be accepted without revision

Review form: Reviewer 2

Is the manuscript scientifically sound in its present form?

Yes

Are the interpretations and conclusions justified by the results?

Yes

Is the language acceptable?

Yes

Do you have any ethical concerns with this paper?

No

Have you any concerns about statistical analyses in this paper?

No

Recommendation?

Accept with minor revision (please list in comments)

Comments to the Author(s)

Comments have been addressed. Hence, acceptance is recommended.

Decision letter (RSOS-200378.R1)

Dear Dr Yang,

It is a pleasure to accept your manuscript entitled "Janus microdimer swimming in oscillating magnetic field" in its current form for publication in Royal Society Open Science. The comments of the reviewers who reviewed your manuscript are included at the foot of this letter.

on behalf of Professor Miles Padgett (Associate Editor) and Pietro Cicuta (Subject Editor)
openscience@royalsociety.org

Reviewer comments to Author:

Reviewer: 1
Comments to the Author(s)

This Paper can be accepted without revision

Reviewer: 2

Comments to the Author(s)

Comments have been addressed. Hence, acceptance is recommended.

Appendix A

RESPONSE TO EDITORS

Thank you for your kind suggestion. The written English of manuscript have been edited by the language editing service which Royal Society recommended.

This document certifies that the manuscript
Janus microdimer swimming in oscillating magnetic field
prepared by the authors
Jinyou Yang
was edited for proper English language, grammar, punctuation, spelling, and overall style
by one or more of the highly qualified native English speaking editors at AJE.
This certificate was issued on **September 18, 2020** and may be verified
on the AJE website using the verification code **88F3-4A0C-5F97-5863-69AB**.

Neither the research content nor the authors' intentions were altered in any way during the editing process. Documents receiving this certification should be English-ready for publication; however, the author has the ability to accept or reject our suggestions and changes. To verify the final AJE edited version, please visit our verification page at aje.com/certificate. If you have any questions or concerns about this edited document, please contact AJE at support@aje.com.

AJE provides a range of editing, translation, and manuscript services for researchers and publishers around the world. For more information about our company, services, and partner discounts, please visit aje.com.

RESPONSE TO REFEREE 1

We thank the reviewer for the time you took to review our manuscript, and many kind suggestions to improve our paper. We have sought to address each and every point raised by the reviewer with the utmost care. We believe that the manuscript is considerably improved as a consequence.

Below is a point-by-point response to the reviewer. The comment from the reviewer is presented in **blue color**, and the answer is presented in black color. In the main manuscript, the revised parts are highlighted in red.

Comment 1: have you verified the numerical model? Is it enough for numerical simulation with 320 elements on the janus particle?

Reply 1: The equations and numerical model in the manuscript are compared to references. And the boundary element method was used in our former study and widely verified by similar research. The number of element was verified that was good enough between the accuracy and the CPU time cost. And we compared the simulation results and the experiment results [Li et al.2018]. Therefore, we used the numerical model with 320 elements for simulation.

Comment 2: Have you verified experimentally the method of microrobots propelled by two orientation oscillating magnetic field without wall?

Reply 3: We are sorry, we have not experimentally verified this part, because there are no related devices for this experiment in our lab.

Comment 3: Please cite and discuss the latest work of microrobots in introduction, such as Science Robotics, 2019, 4 (32), eaax0613;

Science advances, 2018, 4 (11), eaat4388;

Nanomaterials 2019, 9 (12), 1672;

Advanced Functional Materials, 2018, 28 (25), 1706066;

Nano letters, 2017, 17 (8), 5092-5098;

ACS nano, 2017, 11 (9), 9268-9275.

Reply 3: Thank you for your thoughtful suggestions. We modified the Introduction and referred the suggested papers. They are also included in References.

Comment 4: The English should be polished

Reply 4: Thank you for the suggestions. We sent the manuscript to a language editing service for editing the written English.

RESPONSE TO REFEREE 2

We thank the reviewer for the time you took to review our manuscript, and many kind suggestions to improve our paper. We have sought to address each and every point raised by the reviewer with the utmost care. We believe that the manuscript is considerably improved as a consequence.

Below is a point-by-point response to the reviewer. The comments from the reviewer are presented in blue color, and the answers are presented in black color. In the main manuscript, the revised parts are highlighted in red.

General Comment : This is a fundamental research contribution that analyzes theoretically the motion of a Janus dimer system under oscillating magnetic fields, and the interactions between the dimer and surfaces. The paper is nicely explained.

Reply : We really appreciate the reviewer's kind comments.

Comment 1: However, it is not clear to this author if the surfaces between the microdimers play also a role in this type of locomotion. Can the authors comment on this?

Reply 1: Thank you for your kind comment. In this simulation, we only considered the repulsive force between two particles, and we found the speed of the dimer motion were affected by this force. So the interaction between the surface of microdimers may play also an important role in this motion, and in next, we will some research about this.